

# Ranking, selecting, and prioritising genes with desirability functions

Stanley E. Lazic

In Silico Lead Discovery, Novartis Institutes for Biomedical Research, Basel, Switzerland

## ABSTRACT

In functional genomics experiments, researchers often select genes to follow-up or validate from a long list of differentially expressed genes. Typically, sharp thresholds are used to bin genes into groups such as significant/non-significant or fold change above/below a cut-off value, and *ad hoc* criteria are also used such as favouring well-known genes. Binning, however, is inefficient and does not take the uncertainty of the measurements into account. Furthermore, *p*-values, fold-changes, and other outcomes are treated as equally important, and relevant genes may be overlooked with such an approach. Desirability functions are proposed as a way to integrate multiple selection criteria for ranking, selecting, and prioritising genes. These functions map any variable to a continuous 0–1 scale, where one is maximally desirable and zero is unacceptable. Multiple selection criteria are then combined to provide an overall desirability that is used to rank genes. In addition to *p*-values and fold-changes, further experimental results and information contained in databases can be easily included as criteria. The approach is demonstrated with a breast cancer microarray data set. The functions and an example data set can be found in the desiR package on CRAN (https://cran.r-project.org/web/packages/desiR/) and the development version is available on GitHub (https://github.com/stanlazic/desiR).

## INTRODUCTION

High-throughput biology experiments typically generate long lists of differentially expressed genes, proteins, metabolites, or lipids. This paper focuses on gene expression, but the methods apply equally well to all -omics technologies. After the data are analysed, often the next step is to select a subset of genes for further experiments. While many computational methods have been developed (*Moreau & Tranchevent, 2012*), biologists often manually select genes based on *p*-values, fold-changes, average expression levels, variance across samples, and other criteria. There are, however, several shortcomings with selecting genes in this way. First, hard thresholds are used to dichotomise or bin variables as either expressed/not expressed, significant/not significant, or above/below a fold-change cut-off. Binning continuous variables removes information, introduces bias, and by not taking the uncertainty in the estimated quantities into account, can allocate a gene to the incorrect bin (*Cohen, 1983*; *Streiner, 2002*; *MacCallum et al., 2002*; *Irwin & McClelland, 2003*; *Senn, 2003*; *Owen & Froman, 2005*; *Chen, Cohen & Chen, 2007*; *Lazic, 2008*;

Corresponding author
Stanley E. Lazic,
stan.lazic@cantab.net

*Naggara et al., 2011*; *Bennette & Vickers, 2012*; *Barnwell-Menard, Li & Cohen, 2015*). For example, if a gene is truly expressed, but due to sampling error has an average expression level just below the threshold, it will be removed from further consideration. This problem is compounded when multiple hard thresholds from several criteria are combined. Second, all selection criteria are implicitly given equal importance, which is often unsuitable. For example, fold-changes might be considered more relevant than *p*-values in a pilot study or an underpowered experiment. Finally, there may still be thousands of genes that meet the chosen criteria and there is no clear way to sort or reduce the list further: genes with the smallest *p*-values are not necessarily the best candidates to follow-up. In such cases researchers may turn to informal methods such as selecting genes that are known to be in relevant pathways or are recognised from a recent publication. The combination of multiple hard thresholds and *ad hoc* methods can miss potentially interesting genes.

Desirability functions are proposed for ranking, selecting, and prioritising genes. They were developed in the 1960s by *Harrington (1965)* and later extended by *Derringer & Suich (1980)*. They are frequently used in cheminformatics to rank compounds (*Segall, 2012*; *Bickerton et al., 2012*) and are co-opted here to prioritise genes.

## MATERIALS AND METHODS

Applying desirability functions is a three step procedure: (1) choose the relevant variables to be used as selection criteria, (2) map the values for each variable onto a continuous 0–1 scale using the appropriate desirability function, and (3) calculate the overall desirability as a weighted combination of the individual desirabilities. The genes are then ranked by their overall desirability. These steps are discussed below.

### Choose variables for selection criteria

The biological question will determine the variables chosen as selection criteria, and for most studies will include the fold-change (either uni- or bidirectional) and *p*-value for a comparison of interest. The mean and variance of each gene can be used as nonspecific soft filters to deprioritise genes that are not expressed or have near-constant expression across all samples (*McClintick & Edenberg, 2006*; *Hackstadt & Hess, 2009*; *Bourgon, Gentleman & Huber, 2010*). The real advantage of this approach is that information from databases and other statistical parameters can be used as additional criteria. For example, a translational study using mice might prioritise genes that have a high sequence homology with human genes. Alternatively, genes that are expressed in specific tissues might be prioritised; a study of a neurodegenerative disease may use blood samples, but the main interest is genes that are expressed in the brain. Genes whose protein products are easily accessible and measurable might be preferred for biomarker studies; for example, if they are secreted or located on the cell surface. Genes that are known targets of drugs might be preferred if the intention is to inhibit their function in subsequent experiments using genetic and pharmacological methods.

Additional criteria can be based on complex comparisons of multiple conditions or groups, which are often represented as coefficients in statistical models and their associated *p*-values. Suppose the aim is to fined genes that are differentially expressed in multiple

conditions—such as being up-regulated in Parkinson's, Alzheimer's, and Huntington's disease—compared to controls. The usual hard significance threshold and Venn diagram approach will accumulate false negatives; if the power for each comparison is 0.7, then the power to detect a gene in all three comparisons is only $0.7^3 = 0.343$. Another application is to prioritise genes that are differentially expressed in some conditions but not in others (exclusive OR). For example, genes that are differentially expressed only in Parkinson's disease compared to controls and unchanged in the other two diseases. Another criterion is consistency of expression across groups. Genes with similar expression levels between males and females and those without circadian or seasonal variation might be better biomarkers for a diagnostic assay (*Dopico et al., 2015*). A final example is a dose-response experiment where dose is treated as an ordered categorical factor and analysed with an analysis of variance. The genes of primary interest would be those with a large linear coefficient, but without large quadratic or cubic coefficients.

## Map values to 0–1 with desirability functions

Once the key variables have been selected, the next step is to map their values to a 0–1 scale. The choice of mapping function depends on whether high, low, central, or extreme values are considered desirable. Categorical variables can also be included and examples are shown in Fig. 1. Most functions have user-defined cut points that specify where the functions change. The cut points can be based on relevant values (e.g., $p < 0.05$, fold-change $>2$) or on the properties of the distributions; for example, the top 10% of a distribution gets a maximum desirability. The minimum and maximum desirability can also be chosen, but zero and one are often suitable defaults. In some cases a non-zero minimum desirability might be preferred so that a gene is not excluded. Any function that maps values to a 0–1 scale can be used, but those shown in Fig. 1 should cover most situations in experimental biology.

## Calculate the overall desirability

The criteria are then assigned weights according to their importance. The absolute values of the weights are unimportant, only their relative ratios. For example, values of 20 and 10, 2 and 1, or 1 and 0.5 are equivalent because the first value is twice the value of the second, and therefore will contribute twice as much to the overall desirability. The individual desirabilities and weights are then combined into an overall desirability using a weighted geometric mean (Eq. (1))

$$D = \exp\left(\frac{\sum_{i=1}^{n} w_i \ln d_i}{\sum_{i=1}^{n} w_i}\right), \tag{1}$$

where the natural log of $n$ desirabilities ($d_i, i = 1\ldots n$) are multiplied by the importance weights ($w_i, i = 1\ldots n$), and then divided by the sum of the weights. A geometric mean is used because if any desirability is zero, then the overall desirability is also zero. Thus, if a
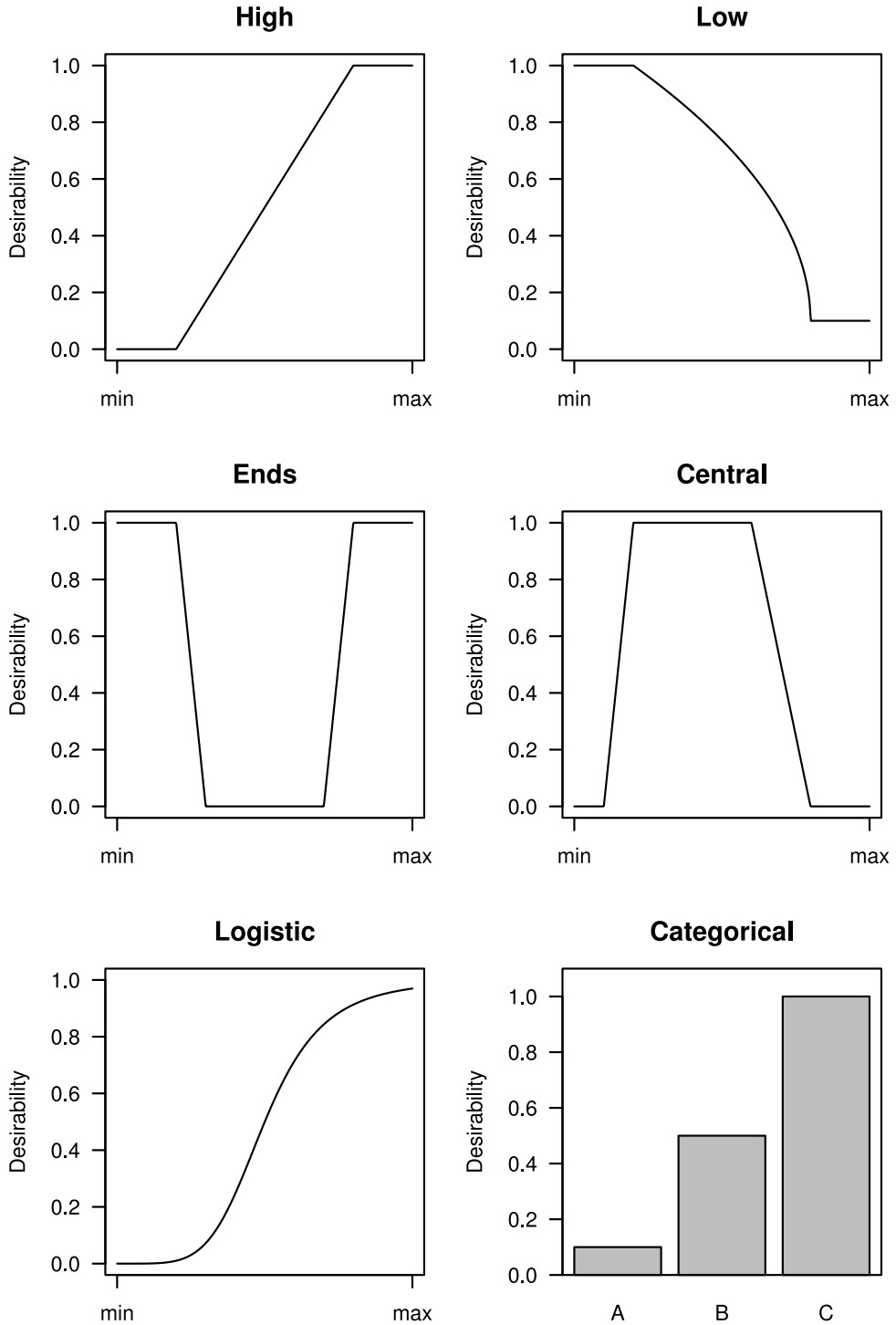

**Figure 1** **Examples of six desirability functions.** Experimental variables (*x*-axis) are mapped to a 0–1 desirability scale (*y*-axis), depending on whether high, low, central, or extreme values are considered desirable. Categorical variables can also be assigned a desirability for each category.

gene is not expressed and has a desirability of zero for this criterion, then the values of the other desirabilities are irrelevant.

One or two main criteria, such as the fold-change and *p*-value, should be given a high weight so that they drive the ranking. Criteria used as soft filters (e.g., average expression) are assigned low weights, and other criteria that are of secondary importance are given intermediate weights. The R script in the Supplemental Information provides a complete analysis for this example.

## RESULTS AND DISCUSSION

### A microarray example

Desirability functions are illustrated with a breast cancer Affymetrix microarray study from *Farmer et al.* (*2005*, GEO: GDS1329). For this example the basal and luminal groups are compared and the apocrine group is not used. Using only a false discovery rate (FDR) cut-off of 0.05, there are 4,830 differentially expressed probe sets between these two groups, which is far too many to follow-up experimentally. Additional variables are therefore used for selection criteria and are shown in Fig. 2. The *p*-value (low values good) and $\log_2$ fold-change (extreme values good) are the two key criteria and receive a maximum weight of one. Probe sets with a *p*-value below 0.0001 are given the maximum desirability ($d = 1$), probe sets with *p*-values greater than 0.1 are completely filtered out ($d = 0$), and probe sets with intermediate *p*-values receive intermediate desirability values. The average expression (high values good) and variability of expression (high values good) are used as soft nonspecific filters and receive a low weight of 0.1—or 10% of the maximal weight. A low weight is used because we do not want the ranked list to be driven by these variables.

*Venet, Dumont & Detours (2011)* showed that randomly selected genes are often good prognostic biomarkers because much of the cancer transcriptome is related to proliferation, and for the purposes of this example, suppose that we are not interested in genes that are mainly markers of proliferation. Each probe sets' correlation with PCNA (proliferating cell nuclear antigen—a marker of proliferation) is therefore used as another criterion to prioritise genes. Probe sets with a low absolute correlation with PCNA are prioritised, and this criterion is given a weight of 0.5, making it half as important as the *p*-value and fold-change. Suppose also that we are interested in inflammatory aspects of the disease. The final criterion therefore is whether a gene is annotated to a gene ontology (GO) term involving inflammation. If so, it receives a desirability of 1, otherwise 0.2. A minimum desirability of 0.2 is used instead of zero because we do not want to remove these genes, only deprioritise them. Also, database information may be incomplete or erroneous and we therefore retain these genes in the final list. A better approach would be to use the GO evidence codes to assign higher desirabilities to annotations that are experimentally verified and lower ones to annotations based on computational predictions. The six criteria are then combined into an overall desirability and a volcano plot highlights the 46 probe sets with overall desirabilities greater than 0.7 (Fig. 3). They are not necessarily those with the smallest *p*-values or most extreme fold-changes but they are the best candidates

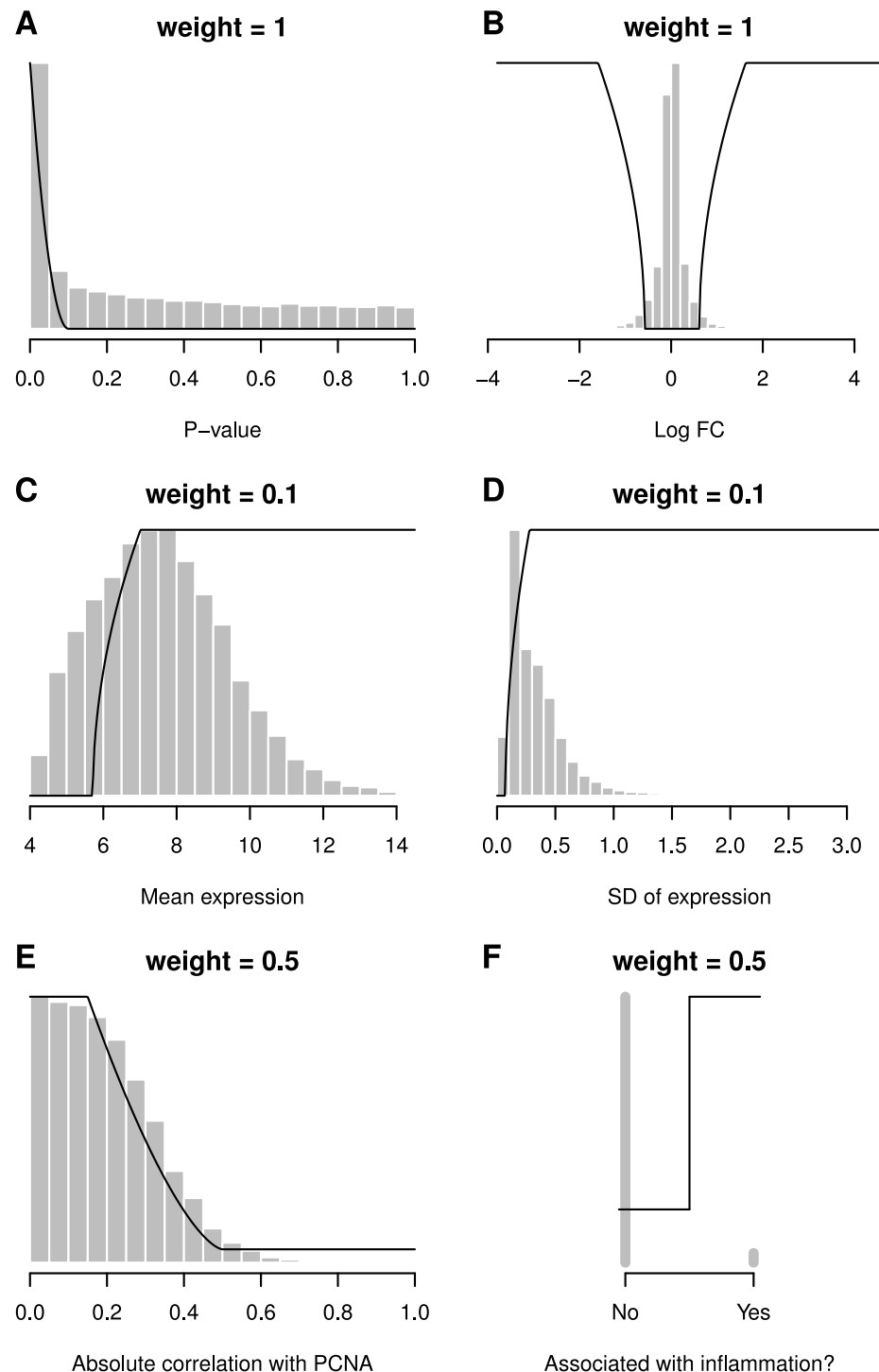

**Figure 2 Desirability functions (black lines), data distributions (histograms) and weights for six criteria.** Small *p*-values (A) and extreme fold-change values (B) are considered desirable. High values for the average (C) and variability in expression (D) are considered desirable, as are low absolute correlations with PCNA (E). Genes annotated to a GO term involving inflammation are given a desirability of 1, otherwise a value of 0.2 (F). *y*-axes are not drawn but range from 0 to 1 for the black lines.

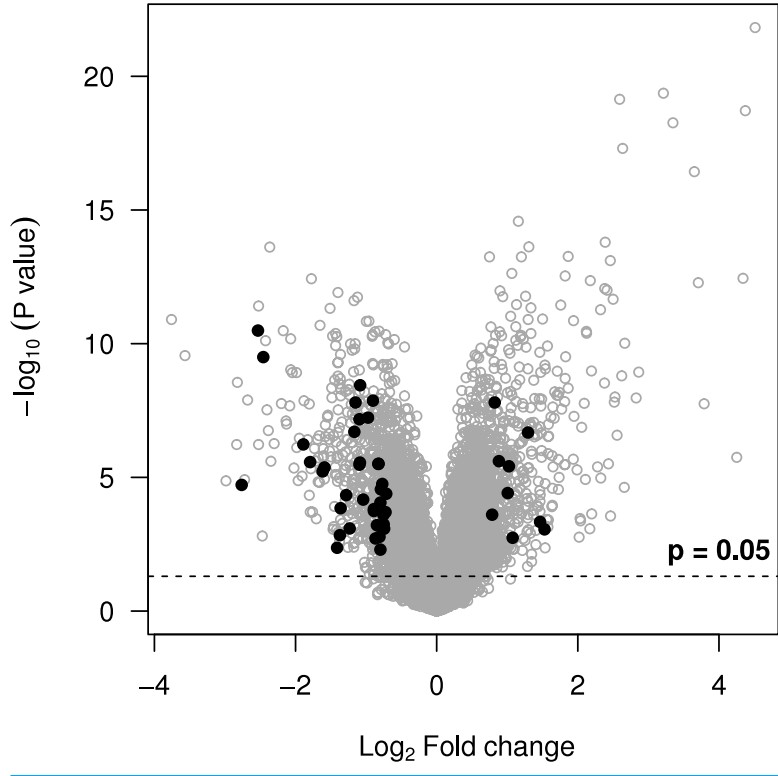

**Figure 3 Volcano plot showing probe sets with an overall desirability greater than 0.7 (black points).** They meet all of the selection criteria but are not necessarily those with the smallest *p*-values or most extreme fold-changes.

given our selection criteria. Alternative desirabilities can be calculated to address different questions or reflect different criteria.

Table 1 shows the top ten probe sets sorted by overall desirability. Most genes are known to be relevant for breast cancer and have an FDR < 0.01, even though the FDR was not a selection criterion. The column *P*-rank shows the probe sets' location in a list sorted by *p*-value (to make the list comparable to a standard analysis, 10,700 low expressed (mean expression <6) and low variance (standard deviation <0.25) probe sets were removed). For example, *S100A9* is ranked 3,168th when the list is sorted by *p*-value and it is unlikely to be selected as a candidate for follow-up experiments, despite having a fold-change of $2^{1.41} = 2.66$, meeting all other criteria, and is involved in breast cancer (*Yin et al., 2013*; *Gumireddy et al., 2014*; *Cormier et al., 2014*). On their own, the results in Table 1 do not prove that these genes are relevant for breast cancer, only that they best met the overall criteria, given all the individual criteria, their mappings, and importance weights. Table 1 therefore represents the beginning of the validation round and not the end of the study. Since the functions can be modified to make favoured genes appear near the top of a list, one should ensure that the ranking reflects the criteria rather than the criteria reflecting a desired ranking. In other words, the workflow is criteria → ranked list → validation, not favourite genes → criteria → scientific claim. As with many methods that have multiple tuning parameters, a sensitivity analysis may be beneficial to determine the degree to

**Table 1 Top ten probe sets sorted by overall desirability.**

| Probeset | Gene | logFC | AveExpr | p-value | P-rank | Overall D |
|----------|------|-------|---------|---------|--------|-----------|
| 202917_s_at | S100A8 | 2.76 | 9.42 | 1.1e−05 | 935 | 1.00 |
| 204470_at | CXCL1 | 1.59 | 6.49 | 1.5e−06 | 668 | 0.95 |
| 214038_at | CCL8 | 1.36 | 8.80 | 1.0e−04 | 1,521 | 0.95 |
| 203535_at | S100A9 | 1.41 | 7.64 | 5.6e−03 | 3,168 | 0.93 |
| 210029_at | IDO1 | 1.23 | 8.90 | 4.5e−04 | 2,249 | 0.92 |
| 209924_at | CCL18 | −1.79 | 9.49 | 2.7e−06 | 594 | 0.92 |
| 32128_at | CCL18 | −1.89 | 9.43 | 5.9e−07 | 440 | 0.91 |
| 206214_at | PLA2G7 | −1.28 | 7.55 | 4.6e−05 | 1,147 | 0.90 |
| 221698_s_at | CLEC7A | −1.09 | 8.15 | 3.4e−06 | 625 | 0.88 |
| 216598_s_at | CCL2 | −1.17 | 8.81 | 2.0e−07 | 347 | 0.87 |

**Notes.**

logFC, log fold-change; AveExpr, average expression; P-rank, rank when gene list is sorted by p-value; Overall D, overall desirability.

which cut points and weights affect the overall results and to check the stability of the final list (*Boulesteix & Slawski, 2009*).

## Integrating data with desirability functions

Desirability functions can not only be used to rank genes within a single experiment, but also to rank genes across several experiments, and to integrate diverse and heterogeneous data—so called data fusion. Suppose a breast cancer proteomics data set with the same two patient groups is available and we are interested in finding genes that are consistently up- or down-regulated in both data sets. An overall desirability can be calculated for the microarray and proteomics data separately, using whatever criteria are deemed relevant. The two experiment-level overall desirabilities can then be combined into a second-level overall desirability, where the weights reflect the experiments' relevance, importance, or quality. This is preferable to hard significance thresholds and Venn diagrams for the reasons discussed above. It may also be preferable to a meta-analysis, since it could argued that the data sets are sufficiently different and are not estimating a common parameter; and with only two experiments, the between-experiment variance will be poorly estimated.

Many methods have been developed and used to combine ordered lists of genes (*Aerts et al., 2006*; *DeConde et al., 2006*; *Kolde et al., 2012*; *Pihur, Datta & Datta, 2008*; *Boulesteix & Slawski, 2009*; *Lin & Ding, 2009*; *Fan et al., 2015*). They are referred to as rank aggregation methods and treat the combination process as an optimisation problem with a large solution space that depends on both the number of genes and the number of experiments. The desirability approach differs in two important ways. First, since rank aggregation converts raw values to ranks, and then normalises the ranks to a 0–1 scale before combining, only a single mapping function is used. This may be useful in many cases but since only one gene can be ranked first, it is not possible to specify that multiple genes are considered equally suitable and should be assigned a maximum ranking. The desirability approach offers greater flexibility with nonlinear mapping functions. Second, since the solution space is extremely large, most studies use only a subset of the ranked genes

(e.g., top 40 genes in each list) to make the analysis manageable. This is another example of binary thresholding, and many genes are excluded even before the data are combined. There are likely many genes that are consistently but moderately differentially expressed across experiments and these can be missed with such an approach. The desirability approach only uses one hard threshold at the very last step, and so all genes are retained until the end. Since the approach is computationally trivial, there is no need to remove genes (mapping the six criteria to a 0–1 scale and combining the results of 22,214 probe sets took less than a second with a 2.93 GHz processor). How the desirability approach compares with the different rank aggregation methods on predictive performance or the ability to select good genes (by some criterion) is still an open question.

There may be missing values when integrating data because an entity may not be measured in all experiments. For example, there may be data for a gene in the microarray experiment but not in the proteomics experiment. In this case the second-level ranking will only use the microarray data. Since the importance weights are normalised to sum to one across all criteria or experiments, criteria with missing values will not influence the ranking, and the remaining criteria will maintain their relative importance. In such cases it is useful to include an additional column in the ranked table indicating the number of experiments or assays for which there are observations.

## What desirability functions are not

Desirability functions are not a machine learning method; they are not expected to select genes whose main virtue is their ability to discriminate between groups or to improve predictions. A gene that is related to proliferation may be an excellent predictor of group membership or survival, but may be irrelevant to a scientist interested in inflammatory aspects of the disease.

Desirability functions are not prior probabilities; they represent preferences for certain outcomes or characteristics over others regardless of their prior probability. Subjective decisions are made, but the same is true for the usual method of selecting genes based on $p$-values and fold-changes, which implicitly use a step-function to map variables to a binary 0/1 scale at a given threshold, and all variables are given equal weight. We might ask whether step-functions are suitable and whether all variables should be given equal weight?

The desirability approach is a simple algorithm that formalises and generalises current methods of selecting genes.

## CONCLUSIONS

Ranking and selecting genes with desirability functions has several advantages. First, a diverse set of criteria can be numerically combined. Second, the importance of the criteria can be formally incorporated into the ranking. Third, non-Gaussian distributions and outliers are not a problem because there is no requirement for a particular distributional form, and extreme data values are constrained to the maximum and minimum desirability values. Fourth, a continuous ranking of genes is returned, rather than just a list of genes that meet all criteria. Fifth, the reproducibility of the analysis is increased because the functions and weights capture the decision criteria. Sixth, the criteria are explicit,

and so can be shared with others, criticised, and modified as needed. Seventh, it is computationally efficient, especially compared to optimisation methods. Finally, the uncertainty in the estimates is taken into account by avoiding unnecessary dichotomisation and mapping estimates to intermediate values between zero and one.

The main disadvantage of this approach is that the results are not probabilistic—there are no *p*-values or confidence intervals associated with the desirabilities. However, the purpose of this approach is not to declare that something new has been discovered, but to select and prioritise genes for further experimentation. A more detailed discussion of the functions and options are provided in the R package vignette (https://cran.r-project.org/web/packages/desiR/vignettes/Gene_ranking.pdf).

## ACKNOWLEDGEMENTS

I would like to thank Pierre Farmer for suggesting the breast cancer microarray data set and for comments on the manuscript, and Ansgar Schuffenhauer and Steffen Renner for comments on the manuscript and/or R package.

### Funding

The author received no funding for this work.

### Competing Interests

The author is an employee of Novartis Institutes for Biomedical Research.

### Author Contributions

- Stanley E. Lazic conceived and designed the experiments, performed the experiments, analyzed the data, contributed reagents/materials/analysis tools, wrote the paper, prepared figures and/or tables, reviewed drafts of the paper.

### Data Availability

https://cran.r-project.org/web/packages/desiR/.

### Supplemental Information

Supplemental information for this article can be found online at http://dx.doi.org/10.7717/peerj.1444#supplemental-information.

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
