# Peer review of "Ranking, selecting, and prioritising genes with desirability functions"

_PeerJ, doi:10.7717/peerj.1444_

## Round 0.1 · original submission · Minor Revisions

As you can see, Reviewer 1 has provided a few suggestions on improving the R implementation, which I hope you will find useful.

The comments by Reviewer 2 are mainly regarding the statistical foundations of your method, and s/he has brought up a number of issues that in my view need addressing. I suggest you pay particular attention to the following points:

1) Do desirability functions reduce misclassification error? In light of Reviewer 2's comments, this claim needs to either be supported by additional evidence or softened/removed.

2) More background to other related techniques needs to be provided.

Reviewer 2 has also provided suggestions on revising some explanatory statements in the Results and Discussion, which I hope you will find constructive.

·

Basic reporting

* The author's email adress is not included on the title page

Experimental design

No Comments

Validity of the findings

No Comments

Additional comments

Review of "Ranking, selecting, and prioritising genes with desirability functions"
by S. E. Lazic
* * *
The author presents a very mature an excellently written manuscript on the selection
of candidate genes for follow up experiments. This is an important,
but largely neglected topic in the analysis of high-throughput data sets in
biology. As the author rightly points out, the criteria commonly used to select candidates
are rather ad-hoc and often have undesirable properties, e.g. accumulation of false
negatives.

The author introduces a very simple and intuitive framework and includes an interesting
case study. Furthermore, the analysis is available as an R-script and the methodology
hase been implemented in an excellently documented R package.


I have just some minor specific comments regarding the manuscript and
the attached script that the author might want to take into consideration.


Comments on the R-script
* * *
1.) Running the R-script in the supplement leads to the error that the
function "d.middle" is not found. From the package vignette, it seem that
the function "d.ends" is needed here.

2.) The direct use of "@" is not considered as a good coding style in R,
rather use the actual getter function, namely use fData(d) and featureData(d)@data


3.) Running

rm.ind <- which(as.character(featureData(d)@data[,"Gene ID"]) == "")
d <- d[-rm.ind, ]

twice will lead to an empty ExpressioSet, since which returns integer(0) the
second time the script is run. (since the condition is always FALSE)

It is considered a good practice
to avoid which in these situations and rather use code like:

rm.ind <- as.character(featureData(d)@data[,"Gene ID"]) == ""
d <- d[!rm.ind, ]

which does not have this problem.


Comments on the manuscript
* * *
4.) Please add a direct link to the package vignette.

Reviewer 2 ·

Basic reporting

No Comments

Experimental design

No Comments

Validity of the findings

No Comments

Additional comments

The manuscript describes an algorithm for rank aggregation using desirability functions in application to prioritization of genes using multiple criteria. I think this approach and the associated software is indeed very useful for any applied researcher seeking to make sense of the often contradicting rankings derived from using alternative ranking criteria (fold changes vs. t-score etc).

While I appreciate the approach as in fact highly useful I think the manuscript itself needs more work.

For example, the approach is motivated in the abstract and in the introduction by stating that "binning is inefficient and creates misclassification error". Unfortunately, any sort of thresholding will lead to type I/II errors, thus the art of determining an optimal threshold consists precisely to minimize these errors. In fact, the aggregated rank list produced by your algorithm also will need to be thresholded to decide which genes are important enough for follow-up study. Thus, this type of argument applies to any list, aggregated or not. Another more valid argument would be that the aggregated ranks are more stable and more robust, and that they integrate diverse information.

Another weakness of the manuscript is the lack of context. There are many methods for aggregating list, both in the general statistical and machine learning literature, as well as in the bioninformatics literature (just search for "rank aggregation" or "aggregation fold-change t-score"). It would be very good if you could place your work in context and at least provide some reference to this earlier work.

When you describe the mapping to desirability functions (in the text and in Fig 1) you state that these provide general mappings to 0-1, but if I look at the figure I recognize that except for the categorical case they are already in on the scale 0-1. Thus, you need to explain a bit better what you are doing here - namely not just mapping to 0-1 (which the statistics apparently already are) but you place weights on specific functional values.

Finally, your discussion of the mehod places a lot of emphasis on what your method is not. I would prefer you see a description what the approach actually is - a simple algorithm formalizing the aggregation of rank list that allows weighting of both the test statistics as well as the individual values. It is thus a very flexible instrument, that in the right skilled hands can produce briiiant results.

Unfortunately, as there are many nobs to turn in your method, you can essentially get any ranking you want by deliberately down/up weighting a method or a specific value. Thus, a word of caution is warranted in your discussion - and the implementation of suitable default settings.

In short, this is a useful method that allows to formalize subjective rank aggregation, and I greatly appreciate that the author provides a corresponding R package.

---

## Round 0.2 · Minor Revisions

Dear Dr Lazic,

Thank you for resubmitting the revised version promptly. I feel that in general your revised version adequately addresses the reviewers' comments, and therefore does not require a re-review. However, in my opinion, some statements introduced in the revision would benefit from more clarity, as indicated in the attached annotated version.

---

## Round 0.3 · accepted · Accept

Thanks for addressing the reviewers' and my comments and apologies for the delay in getting back to you. I also apologise that one highlighted fragment contained no comments - it just meant to indicate that the highlighted statement was somewhat unclear (what exactly could be taken to be an "incorrect bin"?). If you'd like to amend this statement, please feel free to do so in the proof.